# Thermodynamics Evaluation and Verification of High-Sulfur Copper Slag Composite Agglomerate in Oxidation-Roasting-Separation-Leaching Process

**DOI:** 10.3390/ma16010042

**Published:** 2022-12-21

**Authors:** Kai Zhao, Xinghua Zhang, Wei Zhao, Hongwei Guo, Qiaorong Zhang, Changliang Zhen

**Affiliations:** 1School of Metallurgy and Energy, North China University of Science and Technology, Tangshan 063002, China; 2School of Metallurgy, Northeastern University, Shenyang 110819, China; 3School of Iron and Steel, Soochow University, Suzhou 215006, China

**Keywords:** copper slag, oxidation roasting, composite agglomerates, thermodynamics, desulfurization

## Abstract

To solve situation of high-sulfur copper slag utilization, an innovative technology was proposed for oxidation roasting with composite agglomerates. The oxidation-roasting process was studied by factage7.3 software. The thermodynamics were calculated about sulfur removal, sulfur adsorption and decomposition. The adsorption of sulfur oxides and microstructure evolution were analyzed via an oxidation roasting experiment and SEM-EDS. The results show that the matte and As*_x_*S*_y_* in the matrix could be effectively removed via oxidation roasting in an oxygen atmosphere; the sulfoxide was adsorbed to produce CaSO_4_. The Fe_2_SiO_4_ decomposition could be realized at suitable roasting temperature and in an oxygen atmosphere. This is helpful for the magnetic separation of iron and silicon. The thermodynamic and experimental results were preliminarily confirmed to be available in the innovative process.

## 1. Introduction

Most of the world’s copper is produced via pyrometallurgy [1]. The copper smelting methods can be divided into flash smelting (Ottokunp furnace, Inko furnace, etc.) and bath smelting (Essa furnace, Ausmelt furnace, Noranda Furnace, etc.) [2]. It will produce 2.2t copper slag while producing 1t copper [3]. Currently, copper slag is mainly treated by open-air accumulation, and a large amount of accumulation will not only occupy land resources, but also produce environmental pollution [4,5]. The copper slag is rich in valuable elements such as iron, silicon and a small amount of copper, which has high recycling value [6]. In copper slag, the iron and silicon elements exist in a stable fayalite phase. Copper slag also contains a certain amount of harmful elements such as S and As [7], which significantly increases the difficulties of copper slag reutilization.

Copper-slag recovery processes mainly include direct beneficiation, smelting reduction, direct reduction, oxidation modification and leaching. The reduction and iron extraction routes have been widely studied in existing technology. Jing [8] obtained the copper slag pellets via direct reduction–magnetization tests; the metallization rate of the obtained pellet was higher than 92% and the metal recovery reached 89%. Xu [9] prepared the magnetic separation iron powder with iron grade of 91.10% and metallization rate of 94.27% through a high-temperature rapid reduction roasting magnetic separation test. Yang [10] studied the recovery of iron from water-quenched copper slag via a direct reduction magnetic separation method and obtained direct reduced iron powder with an iron grade of 92.05% and recovery rate of 81.01%. Cao [11,12] directly reduced the carbon-containing pellets of copper slag at high temperature to obtain metallized pellets, and obtained iron powder with 90% iron grade and iron recovery rate of 85% via a magnetic separation method. All the above-mentioned studies demonstrated that the reduction route can recover iron from copper slag effectively. However, it should be pointed out that the harmful elements such as S and As will enter the recovered products at high temperature during the reduction of copper slag [13], which not only reduces the quality of recovered iron powder, but also improves the desulfurization pressure of the following steelmaking process [14,15,16,17]. Most harmful content will react with oxygen and escape in the form of gas, which could effectively improve the quality of roasting products [18,19].

Copper slag research is relatively limited in terms of pollution-free processes, and there is no report of industrial applications of the recovery process. Therefore, the copper slag should be treated as clean, providing environmental protection and efficient recovery. A composite agglomerates oxidation roasting process was proposed according to the current utilization status of copper slag resources. Factage7.3 was employed to calculate the thermodynamic parameters of the oxidation roasting process, and the feasibility of the new process was preliminarily verified by experiments and SEM-EDS analysis.

## 2. Materials and Methods

### 2.1. Raw Materials

In this experiment, the copper slag used is from a typical copper smelter, and the chemical compositions are listed in Table 1. It can be seen that the FeO and SiO_2_ contents are 42.74% and 38.01%, respectively. Due to the characteristics of flash smelting, the iron in raw copper ore is completely fused into the slag, which is the reason for the high iron content in copper slag. This indicates that the iron and silicon elements are valuable and necessary to recycle. The copper slag is the by-product of smelting process. Copper was enmeshed in the slag with process of slagging; therefore, there is a small amount of copper in the slag. The sulfur content in the copper slag is 1.19%, which means the desulfurization must be considered in the extraction of iron from copper slag.

Figure 1 shows the SEM-EDS of copper slag. The copper slag is composed of glass phase and small quantities of particles, indicating that the raw material is typical water-quenched slag. The particles are determined as copper mattes via elemental analysis. The components of matte include FeS and CuS, which are distributed in layers. Based on the composition and SED-EDS analysis, the matte is the main occurrence form of sulfur, which is also the main target of desulfurization in the oxidation roasting process.

### 2.2. Thermodynamic Calculation

The Factsage7.3 was employed to calculate the thermodynamics of copper slag during the oxidation and leaching processes. The raw materials, atmosphere and temperature were set by the equlib module to explore the final products of copper matte and fayalite. The reaction module was used for insight into the Gibbs free energy of fayalite oxidative decomposition, matte oxidation and sulfur-oxide adsorption reaction. The phase diagrams of sulfate were drawn to analyze the effects of temperature, sulfur content and oxygen concentration on the occurrence form. Furthermore, the predominance diagram of SiO_2_ alkali leaching was drawn to obtain the appropriate leaching conditions of the copper slag after oxidation roasting.

### 2.3. Experimental Design

Figure 2 presents the detailed experimental steps in the oxidation roasting–separation-leaching process. Firstly, the copper slag was dried at 105 °C for 2 h and crushed to approximately 0.074 mm. Next, the copper slag core pellet was prepared by manual briquetting under 20 MPa. Then, the CaO powder for adsorbing harmful gas was adhered to the surface of copper slag core pellet in a pelletizing disc. After that, the obtained copper slag–CaO composite agglomerate was charged into a muffle furnace and heated to target temperature with a heating rate of 10 °C/min. During the heating process, the air flow was injected into the muffle furnace; steady air flow is conducive to the stability and unity of the experimental conditions. Therefore, the steady air flow was controlled by the flowmeter into the muffle furnace with a flow rate of 300 mL/min. When the sample temperature rose to the target temperature, the samples were taken out from the furnace and cooled to room temperature. After roasting, the copper slag core pellet and CaO coating layer were separated from the composite agglomerates by a vibrating screen. The obtained copper slag core pellets were crushed for leaching in alkali solution. The obtained CaO coating layer can be recycled as a cement raw material.

## 3. Results

### 3.1. Thermodynamics of Sulfur Oxidation

The sulfur-containing substances in the copper slag mainly includes FeS, CuS, As*_x_*S*_y_* and sulfate. Theoretically, the oxidation and decomposition reactions of these sulfur-containing substances can occur at appropriate roasting temperatures and oxygen partial pressure. In this study, the thermodynamics of sulfur oxidation and decomposition were calculated by Factsage [20] to evaluate the feasibility of sulfur removal from copper slag during the oxidation roasting process. In the calculation, the initial FeS or CuS content is set as 1 mol, the oxygen partial pressure is P(O_2_) = 21% and the temperature ranges from 300 °C to 1200 °C. Figure 3a shows the equilibrium of FeS and O_2_. It can be seen that the FeS transforms into Fe_2_O_3_ and Fe_2_(SO_4_)_3_ with a roasting temperature lower than 700 °C. When the temperature exceeds 700 °C, the Fe_2_(SO_4_)_3_ starts to decompose and transforms into Fe_2_O_3_. Simultaneously, the sulfur volatilizes in the form of SO_2_ and SO_3_. This indicates that the FeS can react with oxygen to produce Fe_2_O_3_ and realize the sulfur removal in the form of gaseous sulfur oxides when the roasting temperature is higher than 700 °C. Figure 3b shows the equilibrium of CuS and O_2_. Clearly, the CuS reacts with oxygen directly to form CuSO_4_ at a low temperature. When the roasting temperature reaches 800 °C, CuSO_4_ begins to decompose to form CuO and gaseous sulfur oxides. This also manifests that the original CuSO_4_ in copper slag will also transform into CuO. When the roasting temperature reaches 900 °C, CuS is completely converted to CuO. When the temperature exceeds 1000 °C, CuO will begin to convert to Cu_2_O. This change in the copper’s form cannot affect the sulfur oxidative removal. In conclusion, the sulfur in the matte can be removed in the form of gaseous sulfur oxide when the oxidation temperature reaches 900 °C.

The sulfur also existed in the form of As*_x_*S*_y_* in copper slag, which has been demonstrated previously in the literature [21,22]. The forms of As*_x_*S*_y_* in copper slag include AsS, As_2_S_2_, As_2_S_3_, As_4_S_4_ and As_4_S_6_. In this calculation, the content of S in As*_x_*S*_y_* is set as 2 mol, the oxygen partial pressure is P(O_2_) = 1%, 6%, 11%, 16%, 21% and the temperature ranges from 300 °C to 1200 °C. The obtained results are summarized in Figure 4. Clearly, most of the sulfur can be oxidized to SO_3_ at low temperature. With the increase in roasting temperature, the proportion of SO_3_ decreases, while the proportion of SO_2_ increases gradually. This is due to the fact that the SO_3_ is unstable at high temperature and tends to decompose. At constant roasting temperature, the proportion of SO_3_ is increased by enhancing the oxygen partial pressure, which is because the high oxygen partial pressure can inhibit the SO_3_ decomposition reaction. This indicates that low oxygen partial pressure is conducive to the preformation of SO_2_. The influence of oxygen partial pressure on the products’ formation can be reflected by the difference between the maximum and minimum SO_2_ values at the same temperature. Clearly, the influence is significant when the oxidation temperature ranges from 500 °C to 900 °C, indicating that the oxygen partial pressure significantly affects the sulfur oxides’ formation proportion. The abovementioned discussion manifests the oxidation of arsenic sulfide, which can easily occur in aerobic conditions, and the sulfur could escape in the form of gaseous sulfur oxides by controlling the temperature and oxygen partial pressure.

The sulfate in the copper slag mainly exists in the form of CaSO_4_, CuSO_4_, Fe_2_(SO_4_)_3_ and MgSO_4_. The Phase digram module is used to calculate the dominant region of the four sulfates. By setting the pressure of the reaction as normal pressure, the Y-axis as logarithm of oxygen pressure, and X as temperature, the phase diagram could be obtained for the dominant region of sulfate. It calculates the effect of the proportion of different oxygen pressure and the temperature on the sulfate equilibrium product under normal pressure.

The predominance diagrams of Ca-S-O_2_, Cu-S-O_2_, Fe-S-O_2_ and Mg-S-O_2_ are presented in Figure 5. In the Ca-S-O_2_ system, the phase exists in the form of CaSO_4_ at low temperature, and CaSO_4_ decomposes to CaO with increasing temperature. Actually, the decomposition temperature of CaSO_4_ is significantly affected by the oxygen partial pressure. The higher the oxygen partial pressure, the higher the decomposition temperature of CaSO_4_. In air atmosphere, the decomposition temperature of CaSO_4_ is about 1120 °C. In the Cu-S-O_2_ system, the phase is CuSO_4_ at low temperature. With increasing temperature, CuSO_4_ could be transformed to CuO and CuSO_4_ composite phases when the oxygen potential is high. The CuO appears in the air atmosphere when the temperature is higher than 700 °C. In the Fe-S-O_2_ system, iron exists in the form of Fe_2_(SO_4_)_3_ at low temperature. When the oxygen partial pressure P(O_2_) is 10^−3.5^, Fe_2_(SO_4_)_3_ is transformed to FeSO_4_ when the temperature increases. With the temperature continuously rising, Fe_2_(SO_4_)_3_ and FeSO_4_ will be decomposed to sulfur oxide and iron oxide. In the Mg-S-O_2_ system, the phase is MgSO_4_ at low temperature. When the temperature increases, MgSO_4_ transforms to MgO and releases sulfur oxides. In conclusion, the CuSO_4_, Fe_2_(SO_4_)_3_ and MgSO_4_ could decompose and volatilize sulfur oxide gas in air atmosphere when the roasting temperature reaches 1000 °C. Notably, the decomposition temperature of CaSO_4_ is higher than the other three sulfates, which also demonstrates the feasibility of selecting CaO to adsorb gaseous sulfur oxide in the oxidation roasting process.

To ensure the environmental protection of the experiment and subsequent process, calcium oxide was selected as the sulfur oxide adsorbent. The thermodynamic calculation of sulfur oxide and calcium oxide is shown in Figure 6. As shown in Figure 6a, the Gibbs free-energy curve indicates that the reactions between sulfur oxide and calcium oxide adsorption can theoretically occur in the roasting temperature range. Figure 6b shows the equilibrium phase diagram of the adsorption process, indicating that the final adsorption product is CaSO_4_. The thermodynamic calculation results shows that the sulfur oxides escaping from the copper slag in the oxidation roasting process can be adsorbed and consolidated by the calcium oxide layer, which alleviates the gaseous sulfur oxides emission and improves the cleanliness of the oxidation-roasting process.

### 3.2. Thermodynamics of Fe_2_SiO_4_ Decomposition and SiO_2_ Leaching

The Fe and Si mainly existed in the form of fayalite (Fe_2_SiO_4_) in the copper slag. The decomposition of fayalite could realize the separation of Fe and Si in the copper slag. The Gibbs free energy and equilibrium phase of fayalite decomposition were calculated using the equlib and reaction module. As shown in Figure 7a, the Gibbs free energy of two oxidative decomposition reactions is much lower than zero, indicating that there are two possible reactions. As shown in Figure 7a, there are two processes for the decomposition of fayalite in air atmosphere; the products reacted with oxygen are Fe_2_O_3_ or Fe_3_O_4_. The Gibbs free energy of two oxidative decomposition reactions is much less than 0, indicating that the oxidative decomposition progress is easy to carry out. Figure 7b shows the equilibrium phase diagram of the fayalite oxidative decomposition. In this calculation, the initial amount of Fe_2_SiO_4_ was set as 1 mol, the oxygen addition amount was set as 0.1 mol/step and the temperature was set at 1000 °C. The results show that the Fe_2_SiO_4_ gradually decomposes into Fe_3_O_4_ and SiO_2_ at 1000 °C. With the addition of oxygen, the fayalite continues to decompose, and Fe_3_O_4_ begins to transform to Fe_2_O_3_. The final products are SiO_2_ and Fe_2_O_3_ when oxygen addition is sufficient. This indicates that the oxidation decomposition conditions of fayalite cooperates with the sulfur removal conditions of copper slag.

In order to separate and recover the Fe and iron from the roasted copper slag, alkali leaching of the core pellet was conducted. In the alkali leaching thermodynamics calculation, the degradation amount of SiO_2_ was set at 1 mol/kg, oxygen partial pressure was set as the X- axis, and NaOH addition was set as the Y-axis. As shown in Figure 8, when NaOH is added in a small amount, the equilibrium system consists of SiO_2_ and NaOH solution, and there is no reaction at equilibrium; notably, the neglect of time in thermodynamics leads to the formation of H_4_SiO_4_ in aqueous solution, which is not found in the actual experiment. With the NaOH gradual addition, part of Si reacts with it to form Na_2_Si_2_O_5_, which is completely transformed when the addition amount is 1.5 mol/kg. After addition of NaOH, the reaction products are divided into Na_2_SiO_3_ and Na_2_Si_2_O_5_, and the constituents tend towards equilibrium due to the increase in sodium ion concentration. When the amount of NaOH is greater than 2.5 mol/kg, SiO_2_ is completely transformed into Na_2_SiO_3_. Thermodynamic analysis shows that the silica-rich phase in roasting copper slag could be leached out by NaOH solution with a certain concentration, the iron-rich phase enters the tail slag, and the silica-rich phase is completely transformed into Na_2_SiO_3_, thus realizing the iron–silicon separation of roasting products.

### 3.3. Desulfurization and Adsorption Behavior in Oxidation-Roasting Process

To verify the feasibility of thermodynamic analysis, the desulfurization and solidification experiments of high-sulfur copper slag agglomerate was carried out. The experiment proceeded in a ventilated tubular furnace. The experimental sample was heated in an air atmosphere and the heating rate was 10 °C/min. The copper slag sample was brought out when the constant temperature zone reached 1000 °C. After cooling to room temperature, the interface of adsorbent was analyzed by SEM and EDS. The interface scanning image of the experimental sample is shown in Figure 9.

It could be found from the line scanning images that the Ca, Si and Fe cannot obviously migrate and distribute on both sides of the interface after roasting at high temperature. However, the content of S obviously increased at the interface, indicating that sulfur was absorbed in large quantities. The copper slag oxidation roasting is a gas–solid reaction; the sulfur was adsorbed and consolidated by the CaO coating layer when sulfur oxide gas volatilized. The elemental distribution images shows that the Si and Fe coincided in the copper slag area, the Ca existed in the adsorbent area and the S had obvious aggregation at the interface. This could be divided into two parts for discussion. The absence of sulfur in the copper slag layer is due to the large amount of sulfur oxidation. The area where sulfur accumulates is the area closest to the copper slag layer. The CaO layer has obvious adsorption behavior of sulfur and the process is completed at the interface; therefore, sulfur is not detected at the upper part of the adsorption layer. Notably, the two distribution images of S and Ca elements perform obvious overlapping parts. According to thermodynamic analysis and experimental results, the consolidation product is CaSO_4_.

The XRD patterns of the core pellet after oxidization at different roasting temperatures are shown in Figure 10. The main components of initial copper slag are Fe_2_SiO_4_ and a small amount of Fe_3_O_4_ at room temperature. With the increase in roasting temperature, the Fe_2_SiO_4_ gradually transformed to Fe_2_O_3_, and the intensity of Fe_2_O_3_ diffraction peaks gradually improved with the temperature increase, reaching its strongest point at 1000 °C. This indicates that the higher temperature benefits the oxidation and decomposition of Fe_2_SiO_4_. In addition, there is a side peak of Fe_3_O_4_ on the left side of the main Fe_2_O_3_ peak at about 37°. The existence of Fe_3_O_4_ indicates that the oxidation and decomposition of copper slag is not complete due to the insufficient temperature and time. The silica-containing products cannot be detected in the patterns, which is because the SiO_2_ decomposed by Fe_2_SiO_4_ exists in amorphous crystalline form.

To further determine the phase composition in copper slag after oxidation roasting, the FTIR analysis was conducted. Figure 11 shows the infrared spectrum of copper slag samples oxidized at different temperatures. At 600 °C, the silica absorption peaks mainly appeared at 462 cm^−1^, 804 cm^−1^, 862 cm^−1^, 949 cm^−1^ and 1081 cm^−1^. The absorption peaks of Fe_2_O_3_ and Fe_3_O_4_ can be observed at 531 cm^−1^ and 591 cm^−1^, respectively. The absorption peak of Fe_3_O_4_ indicates that the oxidation reaction is not complete at 600 °C, which is consistent with the XRD analysis results. With the increase in oxidation temperature, the intensity of the Fe_3_O_4_ absorption peak gradually decreased, indicating that roasting temperature promoted the degree of iron oxidation. It should be pointed out that the absorption peaks at 462 cm^−1^, 804 cm^−1^, 862 cm^−1^, 949 cm^−1^ and 1081 cm^−1^ are different forms of silica, and the silica structure transformed into various spatial structures with the increase in roasting temperature. The FTIR and XRD analysis results proved that the fayalite can be oxidized, and it decomposed iron oxide and amorphous state silica in the roasting process.

### 3.4. Leaching Behavior of Copper Slag in Alkaline Aqueous

The roasted sample was leached in an aqueous solution with a solid/liquid ratio of 5 g/100 mL, duration of 2 h and NaOH concentration of 160 g/L [23]. Then, the precipitation was filtered, and the hydrochloric acid solution was dropped into the aqueous solution. Large amounts of white floc appeared. The collected white floc was further analyzed by SEM-EDS, as shown in Figure 12. Clearly, the precipitated SiO_2_ can be divided into amorphous SiO_2_ and a small amount of dendritic SiO_2_, which explains the SiO_2_ diffraction peak absent in XRD and the existence of two SiO_2_ peaks in FTIR. Thus, it is preliminarily proved that the recovery of silicon and iron from roasted copper slag is available via an alkaline solution leaching method.

## 4. Discussion

In this study, a composite agglomerate with copper slag as the core pellet and CaO as the coating layer was prepared in the laboratory. The gaseous sulfur oxide can be effectively absorbed by the CaO coating layer in the oxidation roasting process. The detailed reaction mechanisms of high-sulfur copper slag composite agglomerates in oxidative roasting are summarized in Figure 13. The original copper slag mainly consists of glass phase and matte composite phase. The main elements in the glass phase are Fe, Si and O, which will precipitate in the form of Fe_2_SiO_4_ during the heating process. In the oxidation roasting process, the matte phases, namely CuS and FeS, are oxidized to generate SO_2_, CuO and Fe_2_O_3_. The escaped gaseous sulfur oxides from the copper slag core pellet can be absorbed by the CaO coating layer, forming a CaSO_4_ layer between the core–shell interface. Simultaneously, the fayalite crystals were separated out in the glass matrix and decomposed to Fe_2_O_3_ and amorphous SiO_2_ during the oxidation process. Finally, the Fe_2_O_3_ and amorphous SiO_2_ can be separated via the alkaline leaching method. Compared with the traditional copper slag treatment process, this study provides an environmentally friendly route to realize the recovery of Fe, Si and S from copper slag; it has lower energy consumption and does not need to add fuel. CaO adsorbed sulfur after using it as a beneficial substance of cement, without wasting resources.

## 5. Conclusions

This study provides an environmentally friendly route to realize the recovery of Fe, Si and S from copper slag. The thermodynamic and experimental results preliminarily verify the feasibility of the copper slag composite agglomerates in an oxidation-roasting–separating leaching process. The obtained conclusions are listed below.

(1) Thermodynamic calculation results show that the FeS and CuS in matte begin to oxidize at 700 °C and 800 °C, respectively, and the As*_x_*S*_y_* can be oxidized to sulfur oxide in air atmosphere. The appropriate oxygen partial pressure P(O_2_) of sulfate decomposition should be higher than 10^−2^.

(2) The main elements in glass phase of copper slag are Fe, Si and O. The fayalite crystals were separated out in the glass matrix and decomposed to Fe_2_O_3_ and amorphous SiO_2_ in the oxidation process. The SiO_2_ could be dissolved from the oxidative copper slag in alkaline solution to obtain iron-rich phase and silicon-rich phase.

(3) In the oxidation roasting process of composite agglomerate, the sulfur in copper slag core pellet could be oxidized to gaseous sulfur oxides and then escaped. The escaped gaseous sulfur oxides can be absorbed by CaO coating layer, forming a CaSO_4_ layer between the core–shell interface.

## Figures and Tables

**Figure 1 materials-16-00042-f001:**
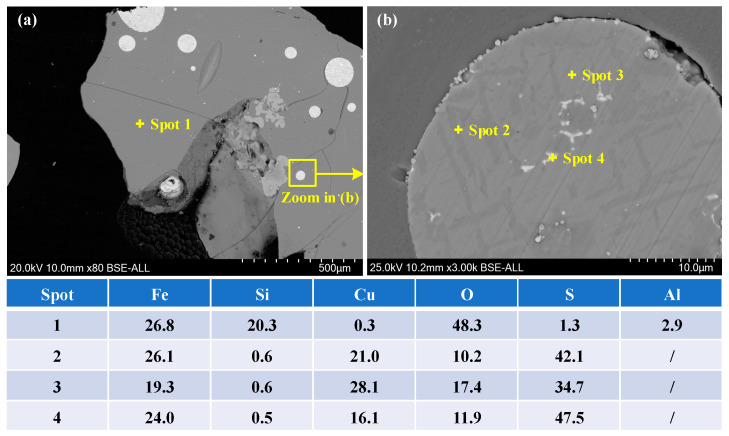
SEM-EDS analysis of copper slag (**a**) SEM in 80 times; (**b**) SEM in 3000 times.

**Figure 2 materials-16-00042-f002:**
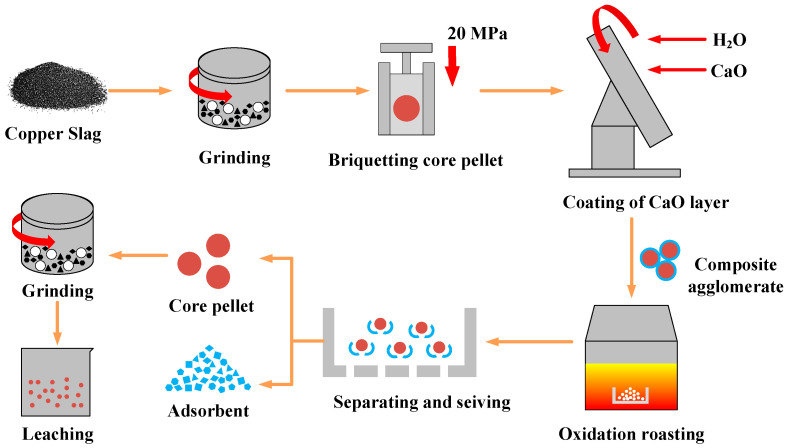
Schematic diagram of oxidation roasting-separation-leaching process.

**Figure 3 materials-16-00042-f003:**
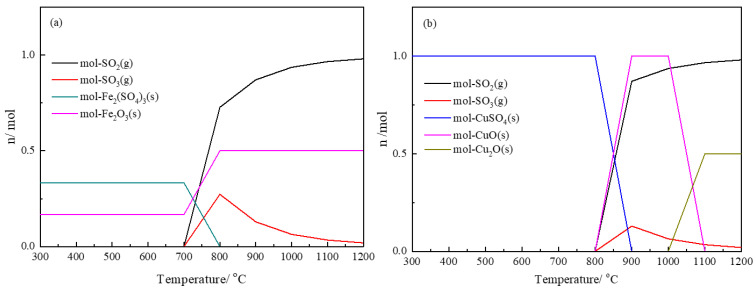
Desulfurization of matte in oxidation roasting (**a**) Oxidation of FeS; (**b**) oxidation of CuS.

**Figure 4 materials-16-00042-f004:**
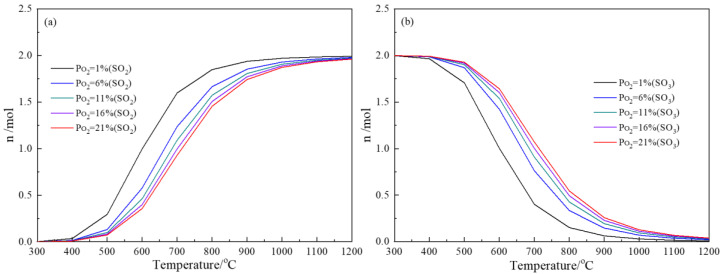
Formation of sulfur oxides in the oxidation of arsenic sulfide under different oxygen partial pressures and temperatures (**a**) SO_2_; (**b**) SO_3_.

**Figure 5 materials-16-00042-f005:**
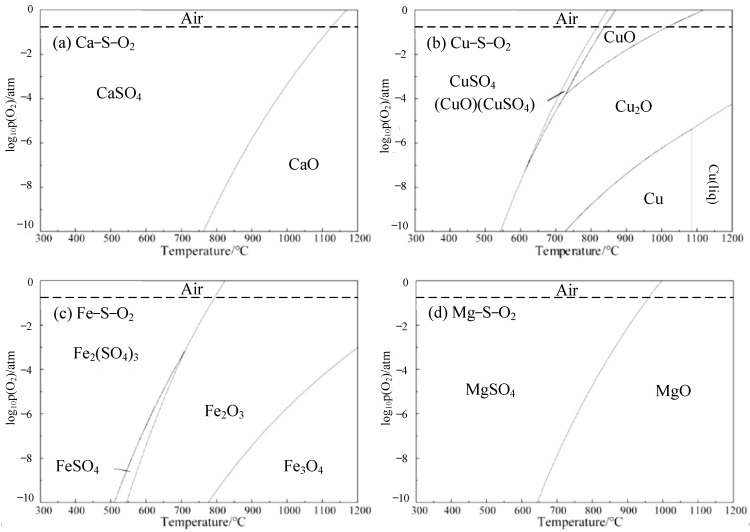
Predominance diagrams of various sulfate desulfurization systems (**a**)Ca−S−O_2_; (**b**)Cu−S−O_2_; (**c**)Fe−S−O_2_; (**d**)Mg−S−O_2_.

**Figure 6 materials-16-00042-f006:**
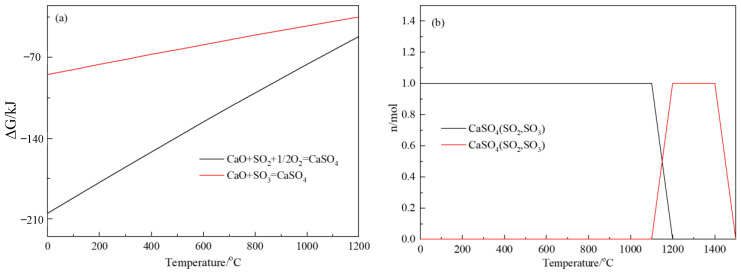
Gibbs free energy (**a**) and equilibrium diagram (**b**) of SO_2_ and SO_3_ adsorption reaction with calcium oxide.

**Figure 7 materials-16-00042-f007:**
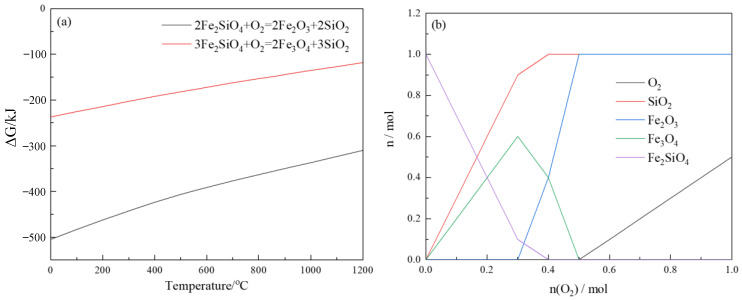
Gibbs free energy curves (**a**) and equilibrium diagram of Fe_2_SiO_4_ decomposition (**b**).

**Figure 8 materials-16-00042-f008:**
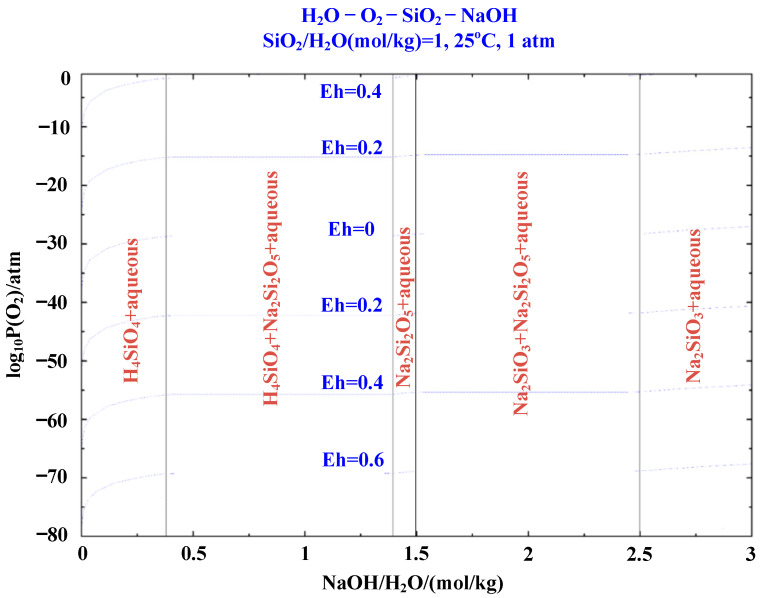
Predominance diagram of SiO_2_ alkali leaching.

**Figure 9 materials-16-00042-f009:**
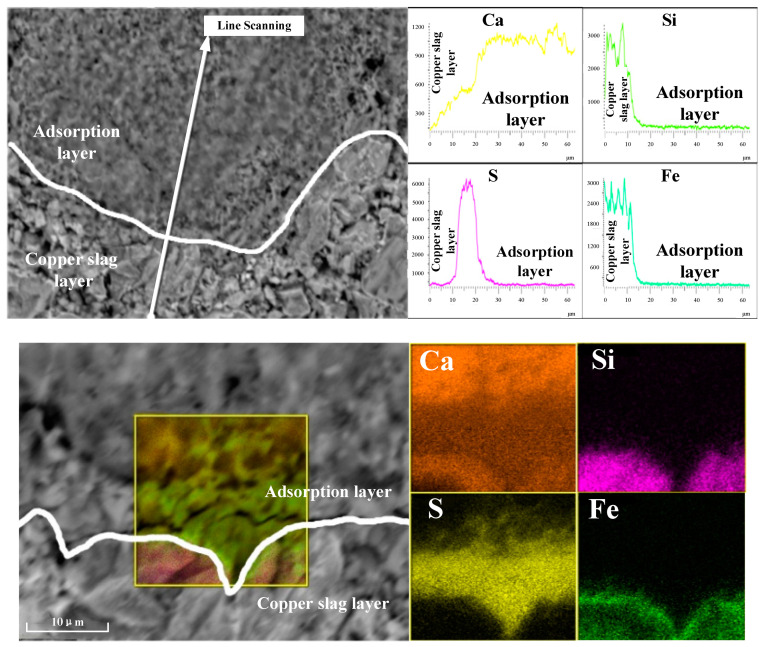
Elemental mapping distribution analysis of the interface between copper slag and calcium oxide.

**Figure 10 materials-16-00042-f010:**
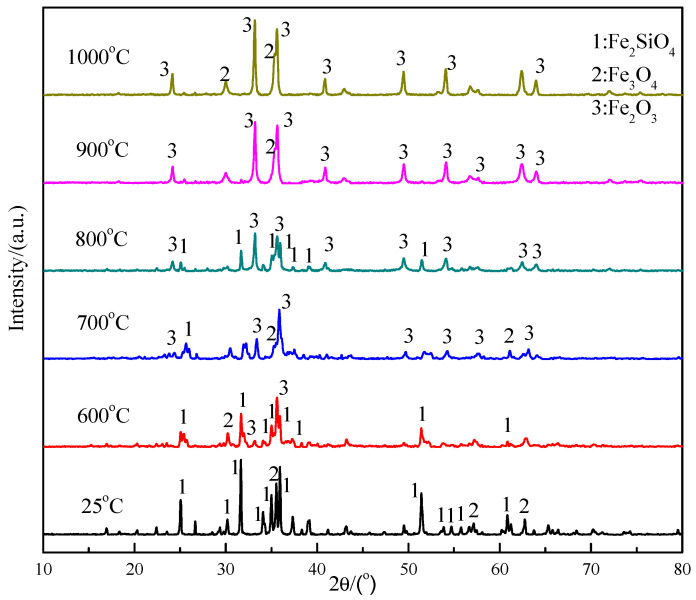
XRD pattern of copper slag in different temperature gradient.

**Figure 11 materials-16-00042-f011:**
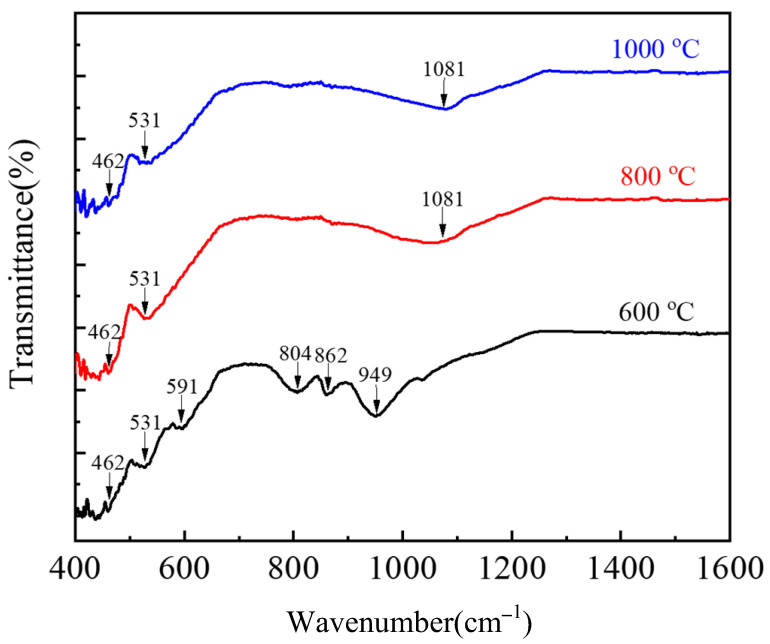
FTIR pattern of copper slag in different temperature gradient.

**Figure 12 materials-16-00042-f012:**
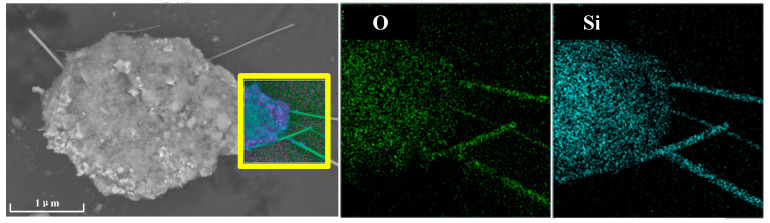
Electron microscope and energy spectrum analysis of white floc.

**Figure 13 materials-16-00042-f013:**
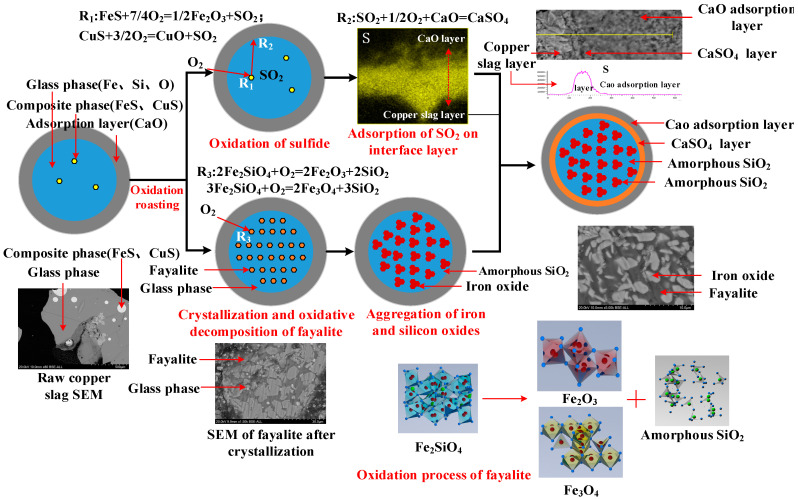
Reaction mechanisms of high-sulfur copper slag composite agglomerate in oxidation-roasting process.

**Table 1 materials-16-00042-t001:** Chemical compositions of copper slag (wt%).

Items	TFe	MFe	FeO	SiO_2_	Al_2_O_3_	MgO	CaO
%	36.10	0.23	42.74	38.01	3.92	1.80	3.41
**Items**	**K_2_O**	**Na_2_O**	**Pb**	**As**	**Cu**	**S**	**P**
%	0.77	0.44	0.31	0.14	0.68	1.19	0.038

## Data Availability

The raw and processed data required to reproduce these results are available by a reasonable request.

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
