# Peer review of "Thermodynamics Evaluation and Verification of High-Sulfur Copper Slag Composite Agglomerate in Oxidation-Roasting-Separation-Leaching Process"

_materials, 2022, doi:10.3390/ma16010042_

Round 1

Reviewer 1 Report

The authors have conducted thermodynamic calculations to explain the high sulfur copper slag utilization. Removal, adsorption, and removal of sulfur have been studied extensively. EDS and some other spectroscopic analyses were done to look at the microstructural features. Here are the comments which the authors should consider:

1.       Fig 9, it is interesting to observe that sulfur is segregated only at the interface of copper slag and adsorption layer. Why do the authors think that such segregation may occur, and sulfur not being found in either of the layers?

2.       Fig 6, the authors should show the Gibbs free energy variation at temperatures above 1200 deg C, and check if they are able to see any cross-over to correlate the equilibrium diagram.

3.       Line 97, In experimental design, what do the authors mean by ‘air flow was introduced into the muffle’? Why have they conducted this step?

4.       Fig 5b, Are the authors sure about the boundary line of Cu and Cu(liq)? Because as you increase the pressure, the stability of Cu is expected to increase. This means that the slope of P-T boundary of Cu/Cu(liq)  should be positive.

5.       Line 221, Pg 8, Remove ‘vertical’ for x-axis.

6.       It will be good if the authors include the construction method of predominance diagrams since these diagrams have been extensively used to explain the observed results.

7.       It is suggested not to put any point symbols rather represent using the lines for the plots which are theoretically/thermodynamic module obtained figures.

Reviewer 2 Report

Dear Authors,

The manuscript is devoted to solving the topical problem of high-sulfur copper slags utilization. The authors proposed an innovative technology of oxidation-calcination roasting with composite agglomerates. The manuscript is well presented and structured. However, there are a number of remarks:

  1. As can be seen from Table 1, the slag contains 0.68% copper. Therefore, slag with such a high metal content should be considered, first of all, as a source of Cu. It is useful to briefly discuss this point.
  2. The authors noted that sulfur is contained in the slag as part of FeS and CuS sulfides. Arsenic is also contained in the sulfide phases. As can be seen from fig. 1, sulfide inclusions are quite large. Grinding the slag down to -74 microns (page 3, line 93) will expose the sulfide spheres. Therefore, using flotation, it is possible to recover copper into a concentrate and at the same time remove harmful impurities - sulfur and arsenic.
  3. It is necessary to discuss the product composition into which arsenic passes during the technology implementation. If arsenic converts to CaSO4, are there any restrictions on the continued use of this product?

In my opinion, taking into account these comments will improve the presented article quality.

Reviewer 3 Report

Reviewer: Minor revision

This paper reports a detailed study about an environmental-friendly route to realize the recovery of Fe, Si, and S from copper slag. The thermodynamic using Factsage7.3 and experimental results preliminarily verify the feasibility of the copper slag composite agglomerates in oxidation roasting-separating leaching process. The proposed approach is very meaningful, and the manuscript organization is satisfied. So, I think that this paper deserves to be published in journal of materials after Minor revision of some issues as follows:

1- Authors must ensure that the quality of English is improved (i.e., make all efforts to rectify any grammatical mistakes, typos, double spaces, missing spaces etc.). For example, in the introduction part line no. 48 abovementioned word must be above mentioned (2 words not one word) also line no. 291 Figure 11 must be correct to Figure 12 and so on.

2- In the result part, the thermodynamics model equations are needed with supported reference(s).  

3- It will be better to compare the results in this work with previously reported works.
